# Fire Safety of Plane Steel Truss according to System Reliability Analysis Combined with FORM Method: The Probabilistic Model and SYSREL Computation

**Katarzyna Kubicka** [1],* and **Milan Sokol** [2]

1 Department of Construction Theory and BIM, Faculty of Civil Engineering and Architecture, Kielce University of Technology, 25-314 Kielce, Poland
2 Department of Structural Mechanics, Faculty of Civil Engineering, Slovak University of Technology in Bratislava, 811 07 Bratislava, Slovakia
* Correspondence: k.kubicka@tu.kielce.pl

**Abstract:** This article focuses on the reliability analysis of the plane steel truss under fire conditions. The safety of the structure was estimated by system reliability analysis combined with First Order Reliability Method (FORM). The authors created the C++ code, which enables us to prepare the advanced probabilistic model for bearing capacity in the selected time of fire duration. Searching cut-sets for system analysis was performed in the C++ code, where stiffness matrix spectral analysis was employed. It was found that a probabilistic model has significant influence on the reliability indices. The research showed that depending on the probabilistic model, the sensitivity of the reliability index to individual variables is different.

**Keywords:** system reliability analysis; First Order Reliability Method (FORM); probabilistic model; fire design situation; steel truss; SYSREL

## 1. Introduction

The paper focuses on the reliability of steel plane trusses under fire conditions with the fully probabilistic model. Therefore, it combines two important issues: structural reliability and design of the structure under fire conditions. The theory of reliability analysis is now well established; there are numerous methods that can be enumerated, including, amongst others: approximation methods (FORM, SORM) [1–4], simulation methods (Importance Sampling, Monte Carlo) [5–8], artificial neural networks [9,10], Stochastic Finite Element Method (SFEM) [11,12], Adaptive Response Surfaces [13,14], or System Reliability Analysis [15–18]. The interesting approach to the management of reliability and predicting the service life of the structure is to use Markov chains [19–21]. All mentioned methods are widely used in many branches of technical sciences to assess the reliability of elements or structures. The reliability index of an element or structure defines its safety and can be assumed to be an alternative to a partial safety factor. The reliability index β is defined as follows:

$$\beta = -\Phi^{-1}\left(P_f\right) \tag{1}$$

where $P_f$—probability of failure,

$\Phi$—Laplace function.

In the paper, the authors computed the reliability of the structure using the SYS-REL program [22], where the system reliability can be combined with approximation or simulation methods.

The part related to obtaining the temperature of the elements and the response of the structure under fire conditions was carried out according to Eurocode [23,24]. Many scientists focus on the reliability aspects of the structure in fire conditions. A few years ago, the probability of failure began to be considered as an alternative to the partial safety

factor for the structure in accidental design situations, when a fully developed fire occurs in the building compartment [25]. One of the interesting but difficult and very wide topics is preparing fire scenarios [26,27]. Another noteworthy problem is the reliability of fire protection coatings [28]. In the following article, neither fire (fire scenarios) nor coating reliability of coatings was considered. It was assumed that the structure is under the influence of a fully developed fire and that the material characteristics of the fire protection coating are deterministic. The response of the structure under fire conditions can be very different depending on many factors. This issue with the assessment of the reliability of steel columns under fire according to the US prescriptive is discussed in [29]. The next important issue is the probabilistic character of the properties of the steel strength [30]. In [31], a comparison of different reliability methods to assess the safety of steel trusses is discussed. Both statically determinate and indeterminate structures are analyzed and the validity of system reliability analysis is demonstrated.

The presented paper focuses on the system reliability of steel structures under fire conditions. In fact, system reliability analysis, searching failure modes, or building failure paths were previously considered in many scientific works. Unfortunately, most of them are related to quite simple structures or are limited to local failure [32,33]. There are some publications about different strategies for finding failure modes in the case of large structures. Shao and Murotsu [34] reviewed typical methods, i.e., the incremental loading method, the branch-and-bound technique, or β-unzipping method. They also proposed a new selective procedure combined with a genetic algorithm. Safari [35] proposed a variant of the non-dominated sorting genetic algorithm used for multi-objective reliability optimization of series-parallel systems. The other method is the gradient tree boosting algorithm, which enables the user to evaluate the safety of steel trusses [36]. To design the structure in a more secure way, the progressive collapse and failure mechanism are willingly modelled in the last scientific papers [37]. Moreover, these methods are useful not only for the structure which is designed, but also for existing ones. It can help, for example, in the diagnosing and preservation of historical bridges [38]. In recent years, reliability methods have been used not only to assess the structural safety in the persistent design situations, but also in accidental situations, including earthquakes, downburst, or fires [39–41]. The last one is the topic that the authors are especially interested in. Thus, the presented article focuses on the system reliability of steel truss under fire conditions. In the presented paper, a fully probabilistic model was proposed and the special tool which enables the analysis with different probabilistic characteristics was created. In previous work, the effectiveness of searching cut-sets by stiffness matrix spectral analysis with a reduced number of elements was proven [41]. Therefore, this method is also used in the following research.

## 2. Materials and Methods

The method presented in the paper connects two main topics: system reliability analysis and design of the structure under fire conditions. The whole analysis was realized in the following steps:

1.  Conducting FEM analysis to obtain the effect of action and bearing capacity of each structure element during fire. This task was accomplished in C++ code, according to Eurocodes.
2.  Finding all possible cut-sets (author-prepared code), i.e., possible ways of transforming the structure into kinematic mechanism. This task is accomplished by spectral analysis of the stiffness matrix, which means that the matrix determinant after exclusion of some set of elements is checked. If it is equal to 0, it means that the structure is geometrically variable, so the mechanism was found.
3.  Estimation of the reliability of single elements. On this basis, knowing the cut-sets, the reliability of the whole structure is estimated using the SYSREL program.

In the following subsections, the single aspects of the research are described in detail.

### 2.1. The Basics of the Fire Design

The basics of designing the structure under fire conditions are described in Eurocode EN 1991-1-2. If the fire behavior is not modeled by specialists in the special software for a specific building, it is allowed to use predefined fire curves. The basic fire curves are as follows: standard, hydrocarbon, and the curve of external fire. If some details of compartments such as floor and walls, and number and area of openings are known, the parametric fire curve can be used. In the presented article, the standard fire curve was applied. In this case, the temperature of fire gases is defined as follows:

$$\theta_g = 20 + 345 log_{10}(8t + 1) \tag{2}$$

where: t—time of fire duration (min).

Knowing the temperature of fire gases, it is possible to calculate the temperature of steel members. For a uniform temperature distribution in a cross section, the temperature increase $\Delta\theta_{a,t}$ of an insulated steel member during the time interval $\Delta t$ is obtained from the following formula [24]:

$$\Delta\theta_{a,t} = \frac{\lambda_p A_p/V \left(\theta_{g,t} - \theta_{a,t}\right)}{d_p c_a \rho_a (1 + \varphi/3)} \Delta t - \left(e^{\varphi/10} - 1\right)\Delta\theta_{g,t} \text{ but } \left(\Delta\theta_{a,t} \geq 0 \text{ if } \Delta\theta_{g,t} = 0\right) \tag{3}$$

with

$$\varphi = \frac{c_p \rho_p}{c_a \rho_a} d_p A_p/V \tag{4}$$

where:

$A_p/V$ is the section factor for steel members insulated by fire protection material, for the cross section with contour insulation, heated from each side, it is defined as the ratio of steel perimeter to steel cross-section area;

Ap is the appropriate area of fire protection material per unit length of the member (m$^2$/m);

$V$ is the volume of the member per unit length (m$^3$/m);

$c_a$ is the temperature-dependent specific heat of steel (J/kgK);

$c_p$ is the temperature-independent specific heat of the fire protection material (J/kgK);

$d_p$ is the thickness of the fire protection material (m);

$\Delta t$ is the time interval; it should not be more than 30 s;

$\theta_{a,t}$ is the steel temperature at time t (°C);

$\theta_{g,t}$ is the ambient gas temperature at time t (°C);

$\Delta\theta_{g,t}$ is the increase in the temperature of the ambient gas during the time interval $\Delta t$;

$\lambda_p$ is the thermal conductivity of the fire protection system (W/mK);

$\rho_a$ is the unit mass of steel (kg/m$^3$);

$\rho_p$ is the unit mass of the fire protection material (kg/m$^3$);

The characteristic of insulation material ($\rho_p$, $c_p$, $\lambda_p$, $d_p$) and unit mass of steel ($\rho_a$) according to Eurocode can be treated as temperature-independent. The dependence of specific heat of steel on temperature is described in detail in part 3.4.1. of the Eurocode [24].

The increase in the temperature of the steel elements results in the decrease of mechanical properties values, including the yield strength and Young's modulus. The appropriate dependencies are presented in Table 1. For the intermediate values of temperature, the linear interpolation can be used.

**Table 1.** Reduction factors for carbon steel at elevated temperatures [24].

| Reduction Factors at Temperature θa Relative to the Value of f<sub>y</sub> or E<sub>a</sub> at 20 °C | | |
|---|---|---|
| Steel Temperature θa [ °C] | Reduction Factor (Relative to f<sub>y</sub>) for Effective Yield Strength $k_{y,\theta}=\frac{f_{y,\theta}}{f_y}$ | Reduction Factor (Relative to E<sub>a</sub>) for the Slop of the Linear Elastic Range. $k_{E,\theta}=\frac{E_{a,\theta}}{E_a}$ |
| 20 | 1.0 | 1.0 |
| 100 | 1.0 | 1.0 |
| 200 | 1.0 | 0.9 |
| 300 | 1.0 | 0.8 |
| 400 | 1.0 | 0.7 |
| 500 | 0.780 | 0.6 |
| 600 | 0.470 | 0.31 |
| 700 | 0.230 | 0.13 |
| 800 | 0.110 | 0.09 |
| 900 | 0.06 | 0.0675 |
| 1000 | 0.04 | 0.045 |
| 1100 | 0.02 | 0.0225 |
| 1200 | 0.00 | 0.00 |

### 2.2. Reliability Analysis

Before estimating the reliability of the whole structure, it is important to compute the reliability of single elements. First, the safety margin *M*, which is a difference between the bearing capacity (*N*) and the effect of action (*E*), should be estimated:

$$M_i = N_i - E_i \tag{5}$$

where index *i* is the number of the element. The subsequent steps include the computation of the following values:

- Standard deviation of safety margin:

$$\sigma_{Mi} = \sqrt{\sigma_{Ni}^2 + \sigma_{Ei}^2} \tag{6}$$

where $\sigma_{Ni}$ and $\sigma_{Ei}$ are standard deviations of bearing capacity and effect of action, respectively.

- Reliability index $t_i$:

$$t_i = \frac{\mu_{Mi}}{\sigma_{Mi}} \tag{7}$$

where $\mu_{Mi} = \mu_{Ni} - \mu_{Ei}$ is the expected value of the safety margin, defined as the difference between expected values of the bearing capacity ($\mu_{Ni}$) and effect of actions ($\mu_{Ei}$).

It is worth noting that reliability index *t* corresponds to an element, while the reliability index *β* refers to a whole structure.

- Probability of failure of an element:

$$P_{f_i} = \Phi(-t_i) \tag{8}$$

- Reliability of the element:

$$R_i = 1 - P_{f_i} \tag{9}$$

It is essential to know reliabilities ($R$) of all elements in order to find reduced cut-sets, i.e., to analyze in the algorithm only these elements, that reliabilities are different from 1. This is easy to explain, when considering Equations (10) and (11). There are two basic structural systems: series (for statically determinate structures) and parallel (for some simply statically indeterminate structures). Most of the statically indeterminate structures correspond to the mixed system, but there are always some combinations of two basic systems: series and parallel ones. Thus, even in these cases, Equations (10) and (11) are useful. Equation (10) corresponds to the series system, the reliability of the whole structure is just a product of reliabilities of all elements. Therefore, it is clear that any *i*-th element, whose reliability is equal to 1 ($Ri = 1$), has no influence on the reliability of the structure.

Equation (11) corresponds to the parallel system. If the reliability of any *i*-th element in this system is equal to 1 ($R_i = 1$), then the expression in brackets equals to 0, so it has no influence on the whole sum in Equation (11)

$$R = \prod_{i=1}^{n} R_i \tag{10}$$

$$R = 1 - \sum_{i=1}^{n}(1 - R_i) \tag{11}$$

*2.3. System Reliability Analysis Combined with FORM*

In the paper, system reliability analysis is computed with the First Order Reliability Method (FORM). Using approximation methods (such as FORM) is essential to transform the limit state function from normal space to Gaussian normal space, which simplifies mathematical operation (because the mean value equals to 1 and standard deviation equals to zero):

$$g_i(\mathbf{X}), \ i = 1, 2, \ldots, m \qquad \rightarrow \qquad G_i(\mathbf{Z}) \ i = 1, 2, \ldots, m \tag{12}$$

In the equation above, $g_i$ and $G_i$ are limit state functions for the *i*-th elements. $\mathbf{X}$ and $\mathbf{Z}$ are vectors of random variables in appropriate spaces: original and standardized, respectively.

All the limit state functions divide the space into two parts: safe and failure areas. For few limit state functions, the total failure area is defined in different ways depending on the system type. In the case of a series system, the total failure area is the sum of individual failure areas (Figure 1a); in the case of a parallel system, this is the common part (Figure 1b).

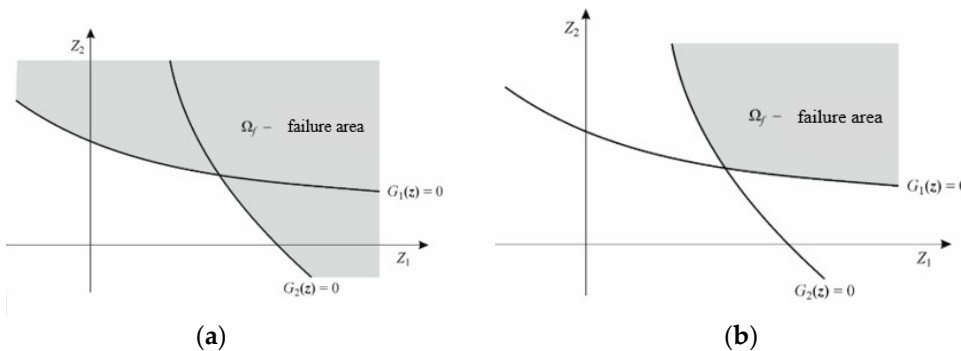

**(a)**           **(b)**

**Figure 1.** Failure areas for: (**a**) series system, (**b**) parallel system.

The failure area ($\Omega_f$) and the probability of failure ($P_f$) of the series system are defined as follows:

$$\Omega_f = \bigcup_{i=1}^{n}\{G_i(\mathbf{Z}) \leq 0\} \tag{13}$$

$$P_f = P\left[\bigcup_{i=1}^{n}\{G_i(\mathbf{Z}) \leq 0\}\right] \tag{14}$$

where: *n*—number of elements of the series system

For the parallel system, these values are defined as follows:

$$\Omega_f = \bigcap_{i=1}^{m}\{G_i(\mathbf{Z}) \leq 0\} \tag{15}$$

$$P_f = P\left[\bigcap_{i=1}^{m}\{G_i(\mathbf{Z}) \le 0\}\right] \tag{16}$$

where: $m$—number of parallel system elements

The failure area of the series-parallel system is defined as follows:

$$\Omega_f = \bigcup_{i=1}^{k}\bigcap_{j\epsilon c_i}\{G_i(\mathbf{Z}) \le 0\} \tag{17}$$

where: $c_i$—the index of $i$-th cut-sets

The equation above reduces to the parallel system formulation when k = 1 and to the series system formulation when each cut-set contains only one component. The probability of failure for the series-parallel system is equal to:

$$\mathrm{P}_f = P\left[\bigcap_{i=1}^{k}\bigcup_{j\epsilon c_i}\{G_i(\mathbf{Z}) \le 0\}\right] \tag{18}$$

For **series systems,** every limit state function $G_i(\mathbf{Z})$ is linearized by Taylor series expansion around the design point with linear coefficient (Figure 2a):

$$\beta_i - \boldsymbol{\alpha}_i\mathbf{Z} = 0 \tag{19}$$

where:

$\beta_i = z_i^* = 0$—the distance from the origin of coordinate system to the design point $z_i^*$,

$\alpha_i = \frac{-\nabla G(z_i^*)}{\nabla G(z_i^*)}$—normalized gradient vector in the design point

Attention should be paid to $\boldsymbol{\alpha}_i$ value, because it allows conducting the analysis according to the FORM method, and also gives information on how sensitive the structure is to changes of any design variable.

In the next step, the new coordinates $Yi\ i = 1, 2, \dots, m$ are introduced. They are defined as the function of $\mathbf{Z} = [Z_1, Z_2, \dots, Z_n]$ coordinates:

$$Y_i = \boldsymbol{\alpha}_i\mathbf{Z} \qquad i = 1, 2, \dots, \mathrm{m} \tag{20}$$

Additionally, the correlation between each pair of $Yi$ and $Yj$ variables is described by the $r_{ij}$ coefficient.

$$r_{ij} = \boldsymbol{\alpha}_i\boldsymbol{\alpha}_j^T, \qquad i = 1, 2, \dots, \mathrm{m}, \qquad j = 1, 2, \dots, \mathrm{m} \tag{21}$$

The element failure is described as follows:

$$F_i = \{\beta_i \le Y_i\} \tag{22}$$

Therefore, the probability of series system failure is calculated according to the following formula:

$$P\left[\bigcup_{i=1}^{m}(\beta_i \le Y_i)\right] = 1 - P\left[\bigcap_{i=1}^{m}(Y_i < \beta_i)\right] = 1 - \Phi_m(\boldsymbol{B}, \boldsymbol{R}) \tag{23}$$

$\boldsymbol{B} = [\beta_1, \beta_2, \dots, \beta_m]^T$

$\boldsymbol{R}$—matrix of correlation coefficients $r_{ij}$.

For the parallel systems, linearization is realized at the common design point, but only for active functions (Figure 2b). Active functions are those limit state functions whose value in the common design point equals zero, so they define the failure area. Functions that have no influence on determination of the failure area are neglected.

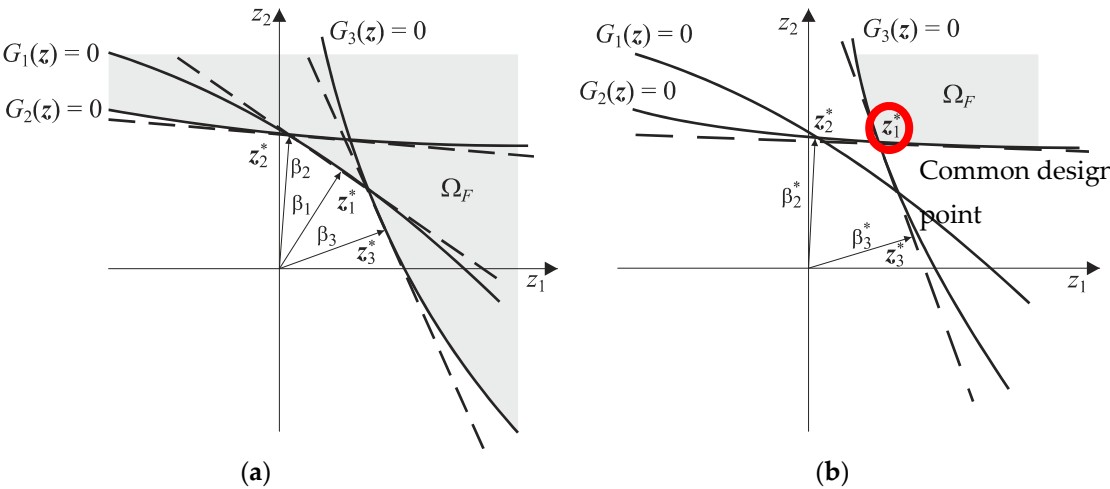

**Figure 2.** Linearization of limit state functions for: (**a**) series system, (**b**) parallel system.

The probability of failure for a parallel system can be computed according to the following formula:

$$P\left[\bigcap_{i=1}^{m_A}(\beta_i^* \le Y_i^*)\right] = P\left[\bigcap_{i=1}^{m_A}(Y_i^* < -\beta_i^*)\right] = \Phi_m(-\boldsymbol{B}^*, \boldsymbol{R}^*) \tag{24}$$

where:

$m_A$—number of active limit state functions,

$\boldsymbol{B}^*$—vector of reliability indices for active functions

$\boldsymbol{R}^*$—matrix of correlation coefficients $r_{ij}$.(only for active functions)

### 2.4. Failure Probability Bounds

Using system reliability analysis, the typical approach is to define the failure probability bounds. This is the way in which the SYSREL program estimates the reliability of the structure. For series system failure, probability bounds are defined as follows [42]:

- First-order bound:

$$\max P(F_i) \le P_f \le \sum_{i=1}^{n} P(F_i) \tag{25}$$

- Second-order bound:

$$P_f = \begin{cases} \ge P(F_1) + \sum_{i=2}^{n} \max\left[0, P(F_i) - \sum_{j=1}^{i-1} P\left(F_i \cap F_j\right)\right] \\ \le \sum_{i=1}^{n} P(F_i) - \sum_{\substack{i=2 \\ j<i}}^{n} \max P\left(F_i \cap F_j\right) \end{cases} \tag{26}$$

In the above, $F$ means failure event, $P_f$ is the probability of failure for the whole system. For parallel system failure, probability bounds are defined as follows [42]:

- First-order bound:

$$0 \le P_f \le \min P(F_i) \tag{27}$$

- Second-order bound:

$$0 \le P_f \le {}_{i,j}^{n} \min\left\{P\left[F_i \cap F_j\right]\right\} \tag{28}$$

### 2.5. Probabilistic Model of the Truss under Fire Conditions

Considering any type of reliability analysis, it is essential to determine the probabilistic model. It has to be decided which variables will be treated as random and which as deterministic. For random variables, the distribution types and coefficient of variation have to be defined. Different probabilistic models lead to getting different values of reliability indices. In previous research, a rather simple model had been used. Let us call this a simplified model. Now, the probabilistic model (advanced model) has been developed. In both models, it was assumed that the truss limit state function for the i-th element ($g_i$) is defined as follows:

$$g_i = N_i - E_i \tag{29}$$

where: $N_i$—bearing capacity of i-th elements, $E_i$—effect of action of i-th element.

In the article, it was assumed that all random variables have normal distribution.

#### 2.5.1. The Simplified Probabilistic Model

In this model, it was assumed that the effect of action (*Eff*) has a coefficient of variation equal to 6%. The authors checked that during the FEM, the coefficient of variation remains practically unchanged. Thus, the coefficient of variation for the effect of action can be equated with the coefficient of variation of external load. The coefficient of variation for the bearing capacity was defined as follows:

- The bearing capacity for the elements in tension ($N_{c,fi}$) and compression with taking into account buckling ($N_{b,fi}$) in the fire design situation is defined as follows:

$$N_{c,fi} = A \cdot k_y \cdot f_y \tag{30}$$

$$N_{b,fi} = \chi_{fi} \cdot A \cdot k_y \cdot f_y \tag{31}$$

- Under assumptions that:
- cross-sectional area ($A$) and yield strength ($f_y$) are random variables with the coefficient of variation equal to $\nu_A = 6\%$ and $\nu_{f_y} = 6\%$, respectively.;
- the other variables in Equations (26) and (27) are deterministic.

The coefficient of variation for the bearing capacity can be estimated according to the following formula [43]:

$$\nu_N = \sqrt{\nu_{fy}^2 + \nu_A^2} = \sqrt{0.06^2 + 0.06^2} = 0.085 = 8.5\% \tag{32}$$

In the simplified probabilistic model, the following values were assumed to be probabilistic: effect of action ($\nu_{Eff} = 6\%$), cross-sectional area ($\nu_A = 6\%$) and the yield strength ($\nu_{f_y} = 6\%$). It was assumed that all these variables have a normal distribution. In such a defined model there is no need either to use any external programs next to SYSREL nor to do any additional computations. The defined coefficients of variation for effect of actions and bearing capacity are just directly introduced to SYSREL. In the article, this easy approach was compared with a more advanced model.

#### 2.5.2. The Advanced Probabilistic Model

Taking into account Equation (27), the bearing capacity of compressed elements, where buckling must be taken into account, depends on many other variables related to buckling coefficient, which is finally the following function:

$$\chi_{fi} = f\left(f_y, \; k_y, k_E, A, E, I_y, \; L\right) \tag{33}$$

Previously (in the case of simplified model), the cross-sectional area ($A$) and yield strength ($f_y$) were defined as random. In fact, they also influence the buckling coefficient, so this value should also be also treated as random value. Furthermore, according to authors, if one of the material characteristics or geometrical characteristics is treated as

a random value, the same should be done with the other values from the appropriate groups. Thus, the simplified probabilistic model was extended by the introduction of the following random variables: Young modulus ($E$) and moment of inertia ($I_y$). The buckling coefficient also depends on the reduction factors ($k_y$, $k_E$), which are functions of the element temperature ($T_a$), which is the function of a few factors:

$$T_a = f(ApV, \text{ insulation characteristics}) \tag{34}$$

where: $ApV$ is the exposition ratio for the elements heated from each side, and it is defined as follows:

$$ApV = \frac{V}{A} \tag{35}$$

In the equation above, $V$ is the perimeter of cross section; this is a consecutive random variable in the advanced probabilistic model. Finally, for this model, the following set of random variables was determined: $f_y$, $E$, $A$, $V$, $I_y$.

Moreover, for the effect of action, the probabilistic model was extended. The effect of the action is a function of following characteristics:

$$Eff = f(T_a, A, V, E) \tag{36}$$

Thus, in the C++ procedure for computation of action effects, a few random variables, including geometric characteristics of cross section ($A$,$V$) and Young modulus ($E$), were introduced. The computation indicated that for such a defined set of random variables, the temperature of elements is deterministic. The authors are aware that this is a very strong assumption, because the fire is a strongly nondeterministic process, and this process is the main source of randomness. Unfortunately, modeling the fire could not be realized easily, and taking into account such an influential factor would probably result in making impossible any comparison of proposed models. The reduction factors ($k_y$ and $k_E$) depend on the temperature element, so it is obvious that they are also random. Furthermore, according to Eurocode they are defined in a deterministic way for a specified temperature. According to the authors, they should be treated as probabilistic, but it is very difficult to find any information about appropriate coefficients of variations.

The sets of all the random variables were generated in a random way from the range determined by the coefficient of variation. The function implemented in C++ generates the variables with homogeneous distribution. It was necessary to transform it into a normal distribution. To achieve this aim, the Box–Müller algorithm [44] was used. Assume that we have two random variables with homogeneous distribution, $U_1$ and $U_2$. Then, it is

$$\Theta = 2\pi U_1 \tag{37}$$

$$R = \sqrt{-2lnU_2} \tag{38}$$

$$X = Rcos\Theta \tag{39}$$

$$Y = Rsin\Theta \tag{40}$$

The derivation is based on a property of a two-dimensional Cartesian system, where the coordinates X and Y are described by two independent and normally distributed random variables, the random variables for $R^2$ and $\Theta$ in the corresponding polar coordinates are also independent.

To make the algorithm of the proposed method easier to understand, it is presented in the graphic form in Figure 3. First, the temperature of fire gases is computed in a deterministic way. The fire curves proposed by Eurocode are the functions that depend only on time of fire duration (t). In this form, it is difficult to take into account any randomness. The authors would like to underline that this is a strong simplification, because the fire is a highly random process. However, fire modeling was not the topic of the article. The temperature of elements was computed with the assumption that the

geometric characteristics of cross-section and the thickness of insulation are deterministic. Then, computations were carried out according to a simplified and advanced model. The probabilistic and deterministic characteristics for both models are presented in Figure 3.

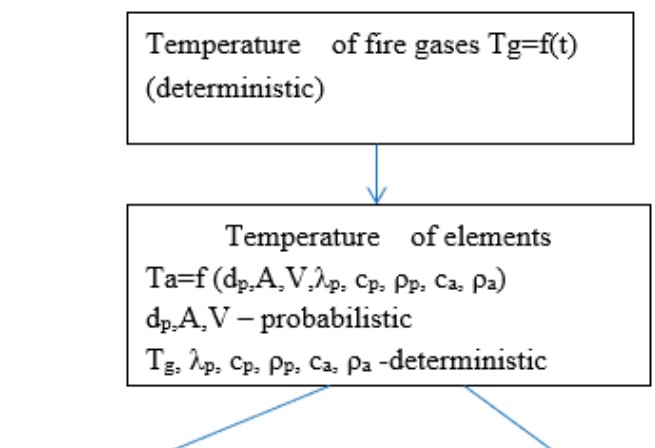

**Figure 3.** The algorithm of computation.{\displaystyle \Theta = 2\pi U_{2}.\,\}.

### 3. Results

Because the article focuses mainly on the description of the methods, a simple example has been chosen. The analysis of the statically indeterminate truss presented in Figure 4 was conducted. The basic assumptions were as follows: the temperature and node forces presented in the figure are the only load, the structure is made of S235 steel, elements 1–12 are made of IPE300, elements 13–25 are made of SHS 60 × 60 × 5, the vermiculite spray-applied fire insulation with a thickness of 2 cm was applied. All results presented below concern the truss in the 30th minute of fire duration.

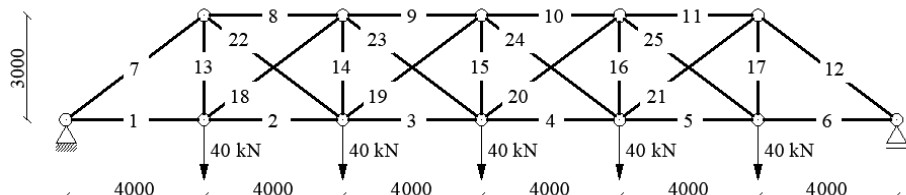

**Figure 4.** The static scheme of the analyzed truss.

With continued fire duration, the effect of action in elements of statically indeterminate truss increases, but the bearing capacity decreases. In the analyzed structure, the bearing capacity of external cross-braces (18 and 25 according to Figure 4) was exceeded before the 30th minute of fire duration. Therefore, it is necessary to consider the new static scheme presented in Figure 5. The digits in the truss scheme correspond to numeration of bars. To

conduct the system reliability analysis, it is essential to find the so-called "cut-sets". In the simplest way, it can be defined as the failures of single elements which lead to transforming the structure into a mechanism. In the paper, stiffness matrix spectral analysis was used to find cut-sets. In this method, all possible combinations of truss elements are removed from the stiffness matrix and its determinant is examined [41,45]. If it is equal to zero, it means the structure is geometrically variable and the cut-set was found. This method is quite time consuming when we check all possibilities, so a good idea is to limit the number of bars, for example by the method proposed in [46]. In this case, all truss elements whose reliability is equal to 1 are neglected. For the analyzed truss, reliabilities of single elements in the 30th minute of fire duration are presented in Appendix A. In Figure 5, elements in red are those bars whose reliability is different to 1, so they were taken into account during searching cut-sets. This task was realized with the usage of C++ code [46]. The searching of cut-sets was realized in the deterministic way, because according to authors, taking into account the randomness would not change the group of neglected elements (which are mostly elements intension).

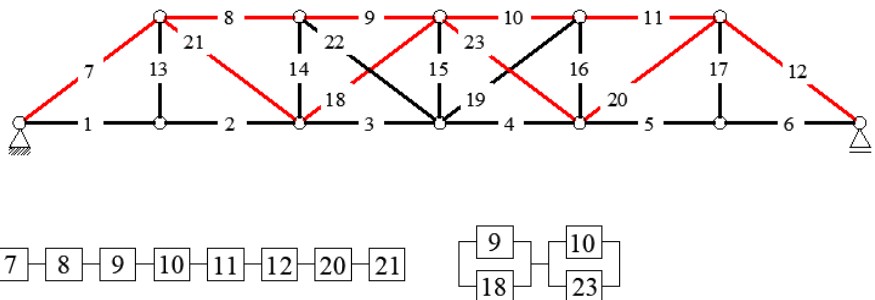

**Figure 5.** The new static scheme of the analyzed truss after 30 minutes of fire duration and corresponding cut-sets.

The analyzed truss in the 30th minute of the duration of the fire can be transformed into a mechanism in two ways (Figure 5). The first corresponds to series systems, which means that failure of any element of the following set: 7, 8, 9, 10, 11, 12, 20, 21 will result in the failure of the whole structure. The second presented system is parallel-series, which means that the failure of the whole structure may be the result of the following elements failures: (9 **AND 18) OR (10 AND** 23).

In the SYSREL program where the system reliability analysis with FORM was performed, the logical model is the cut-sets representation is depicted in Figure 6.

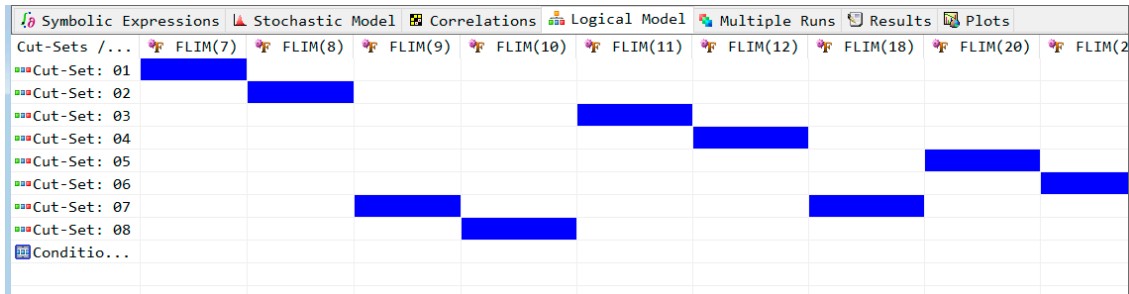

**Figure 6.** Logical model in the SYSREL program.

### 3.1. Results for a Simplified Probabilistic Model

After introducing to the SYSREL program random variables defined according to the simplified probabilistic model, the authors got the results presented in Figure 7. The value of reliability index is marked by red frame. The lower bound is the same as upper bound; this is because the failure probability is very small, so the bounds defined in Section 2.4 are very narrow.

```
*******************************************
------------ Sysrel (Version 12) -------------
----- (C) Copyright: RCP GmbH 1992-2020 ------
*******************************************

--------------------------------------------------------------------
  Job name ............ : bez losowosci
  Comment : No comment
  Transformation type   : Rosenblatt
  Optimization algorithm:    NLPQL
--------------------------------------------------------------------
 *SYSREL*: Linearizing C U T - S E T  No.  1
 *SYSREL*: Linearizing C U T - S E T  No.  2
 *SYSREL*: Linearizing C U T - S E T  No.  3
 *SYSREL*: Linearizing C U T - S E T  No.  4
 *SYSREL*: Linearizing C U T - S E T  No.  5
 *SYSREL*: Linearizing C U T - S E T  No.  6
 *SYSREL*: Linearizing C U T - S E T  No.  7
 *SYSREL*: Linearizing C U T - S E T  No.  8
 Starting evaluation of Ditlevsen Bounds on final union

  Lower Bound Pf, L.B.-beta  ,Upper Bound Pf, U.B.-beta  ,    IER
     7.674E-06     4.324          7.674E-06     4.324            0
 Statistics in SYSREL:
  No.of state-function calls=        70,  gradient calls=    20
  CPU-time overall    0.08, in St.Func.    0.00 seconds
  The error indicator (IER) at the end of SYSREL was =     0

Reliability analysis is finished
```

**Figure 7.** Results of the reliability analysis in the SYSREL program for the simplified model.

The great advantage of the FORM method during the reliability analysis is the opportunity to obtain the values of $\alpha$ coefficient for all random variables. This provides information on how large the influence of any variable changes on the structure reliability is. If $\alpha$ has a positive sign, it means that increasing the corresponding random variable will result in increased reliability. If $\alpha$ is negative, the situation is opposite: an increase in the variable's value causes the decrease of structure reliability. In Figure 8, values of $\alpha$ for all random variables are presented.

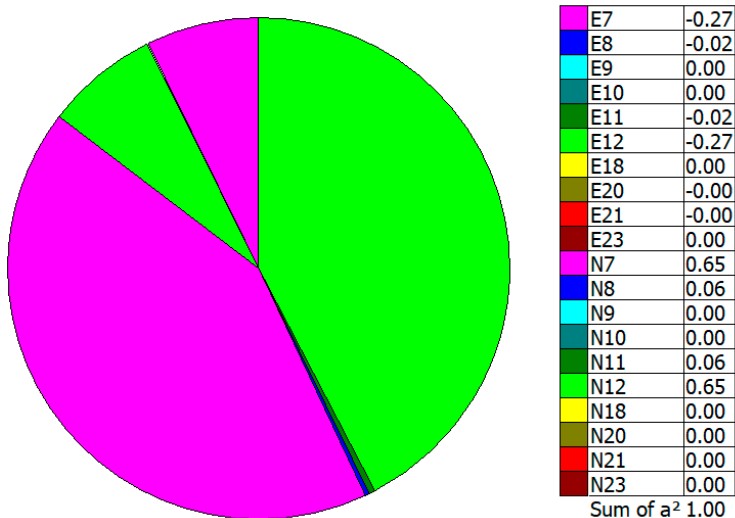

| | E7 | -0.27 |
|---|---|---|
| | E8 | -0.02 |
| | E9 | 0.00 |
| | E10 | 0.00 |
| | E11 | -0.02 |
| | E12 | -0.27 |
| | E18 | 0.00 |
| | E20 | -0.00 |
| | E21 | -0.00 |
| | E23 | 0.00 |
| | N7 | 0.65 |
| | N8 | 0.06 |
| | N9 | 0.00 |
| | N10 | 0.00 |
| | N11 | 0.06 |
| | N12 | 0.65 |
| | N18 | 0.00 |
| | N20 | 0.00 |
| | N21 | 0.00 |
| | N23 | 0.00 |
| | Sum of a² | 1.00 |

**Figure 8.** Representative $\alpha_i$ for random variables (simplified model).

### 3.2. Results for Advanced Probabilistic Model

In the case of the advanced probabilistic model, the coefficient of variation for bearing capacity was computed based on a previously specified set of input random variables. The results are presented in Table 2. Interestingly, the values of temperatures are almost

deterministic during the whole analysis. It is the same situation in the effects of action (see Table 2). The coefficient of variation of effect of action presented in Table 2 has a special meaning. It is the results authors obtained during the computation conducted with C++ with the assumption that external loads are deterministic. It is easy to notice that additional random variables (such as modulus of elasticity or cross sectional area) have no significant influence on the FEM analysis. Therefore, the authors decided that even in the case of the advanced model, the coefficient of variation is the same for the external load and effect of action, i.e., $\nu_E = 6\%$.

**Table 2.** Coefficient in variations of bearing capacities according to advanced probabilistic models.

| Coefficient of Variation (%) | Number of Element | | | | | | | | | |
|---|---|---|---|---|---|---|---|---|---|---|
| | 7 | 8 | 9 | 10 | 11 | 12 | 18 | 20 | 21 | 23 |
| $\nu_{Nb,fi}$ | 7 | 9 | 8 | 8 | 8 | 7 | 7 | 7 | 7 | 7 |
| $\nu_{Nc,fi}$ | 8 | 9 | 9 | 8 | 9 | 8 | 9 | 9 | 9 | 9 |
| $\nu_E$ | 0.1 | 0.1 | 0.2 | 0.3 | 0.02 | 0.1 | 0.2 | 0.8 | 0.3 | 0.4 |

It has to be clearly underlined that the research focused on randomness during the computation. It is obvious that the load applied to the structure is random and that the fire is a highly stochastic process, but it was not the subject of the article. The modeling of fire behavior is a wide and difficult issue. The external load analyzed in the example is just some kind of assumption and should not be identified as any typical type of load. The topic of different types of loads with appropriate distribution types will be developed in further works.

The results of analysis using an advanced probabilistic model are presented in Figure 9. The reliability index is marked in a red frame. The upper and lower bound are the same, because of the reson mentioned in the case of the simplified model. The diagram in Figure 10 shows $\alpha$ values for individual random variables.

```
*******************************************
----------- Sysrel (Version 12) -------------
----- (C) Copyright: RCP GmbH 1992-2020 ------
*******************************************

-----------------------------------------------------------------
 Job name ............. : losowosc poprawione
 Comment : No comment
 Transformation type    : Rosenblatt
 Optimization algorithm:     NLPQL
-----------------------------------------------------------------
 *SYSREL*: Linearizing C U T - S E T  No.  1
 *SYSREL*: Linearizing C U T - S E T  No.  2
 *SYSREL*: Linearizing C U T - S E T  No.  3
 *SYSREL*: Linearizing C U T - S E T  No.  4
 *SYSREL*: Linearizing C U T - S E T  No.  5
 *SYSREL*: Linearizing C U T - S E T  No.  6
 *SYSREL*: Linearizing C U T - S E T  No.  7
 *SYSREL*: Linearizing C U T - S E T  No.  8
 Starting evaluation of Ditlevsen Bounds on final union

   Lower Bound Pf, L.B.-beta ,Upper Bound Pf, U.B.-beta  ,    IER
        2.681E-07     5.013          2.681E-07     5.013       0
 Statistics in SYSREL:
 No.of state-function calls=      70, gradient calls=   20
 CPU-time overall    0.08, in St.Func.   0.00 seconds
 The error indicator (IER) at the end of SYSREL was =   0

Reliability analysis is finished
```

**Figure 9.** Results of the reliability analysis in the SYSREL program for the advanced model.

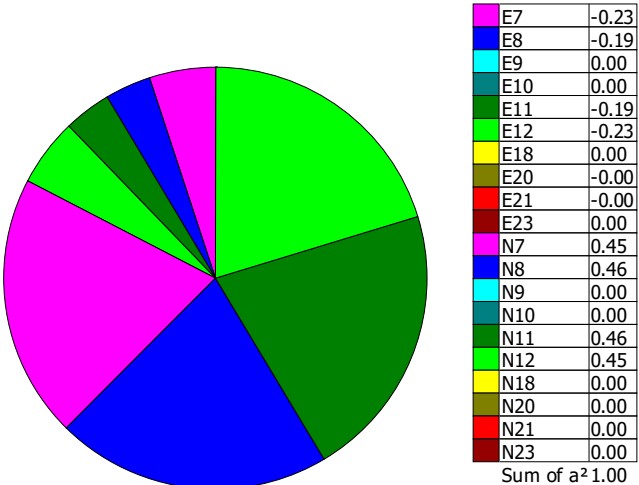

| | | |
|---|---|---|
| | E7 | -0.23 |
| | E8 | -0.19 |
| | E9 | 0.00 |
| | E10 | 0.00 |
| | E11 | -0.19 |
| | E12 | -0.23 |
| | E18 | 0.00 |
| | E20 | -0.00 |
| | E21 | -0.00 |
| | E23 | 0.00 |
| | N7 | 0.45 |
| | N8 | 0.46 |
| | N9 | 0.00 |
| | N10 | 0.00 |
| | N11 | 0.46 |
| | N12 | 0.45 |
| | N18 | 0.00 |
| | N20 | 0.00 |
| | N21 | 0.00 |
| | N23 | 0.00 |
| | Sum of a² | 1.00 |

**Figure 10.** Representative $\alpha_i$ for random variables (advanced model).

## 4. Discussion

The results in case of different values of the reliability indices depend on the probabilistic model. They reach values of 4.324 and 5.013 for the simplified and advanced probabilistic models, respectively. Therefore, it is worth estimating the probabilistic characteristics of the bearing capacity using computer simulation instead of approximation according to Equation (28). In addition, a very interesting fact was found during the sensitivity analysis. Depending on the probabilistic model, the influence of bearing capacities and effects of actions for individual elements on the reliability is different (Figures 7 and 9). It was also found that the input probabilistic variables (cross-section area and Young modulus) have no influence on the character of the effect of action. The process of computing the effects of action under load (including external load as the concentrated forces and forces generated under thermal load) is deterministic, but it should be clearly underlined that the load is highly probabilistic. In the paper, for simplicity of analysis and to make comparison easy, only nodal forces and thermal loads both with normal distributions were assumed. In fact, there are many types of loads, for example, wind, snow, dead load, useful load, etc. Now, it is well known what type of distribution and what characteristic should be used for each of them [47].

## 5. Conclusions

In the article, a specific calculation example was analyzed, but the conclusions have universal character and can be referred to for many other structures, mainly for steel trusses. The results presented in the article indicate unambiguously that the probabilistic model has a significant influence on the results of the reliability analysis. The preparation of the probabilistic model is the biggest challenge and the most difficult task during the safety analysis of a structure. It is not obvious which values should be treated as probabilistic and with which characteristics. This problem is especially evident during the analysis of the structure under fire conditions. The authors of the presented paper managed to solve the problem with the probabilistic character of the buckling coefficient of an element under fire influence. The meaning of the appropriate choice of this coefficient has already been analyzed in the paper [48]. A computer code was prepared that allows the user to obtain the 'real' coefficient of the variation of bearing capacity as a result of computation with any input probabilistic variables. The proposed method is an alternative method to the approximation method using Equation (28). Using the method proposed by the authors, the reliability index is lower, so the results are safer, but it is worth considering if it is not too conservative. It raises a question: Is it possible to change profiles (cross sections) and as a result to reduce the total cost of the structure? The answer is not so easy, because there are still some aspects that need to be analyzed in following work. Especially:

- The trial to conduct the experiments to find out what the true changes of material characteristics in the steel under fire influence are.
- The analysis of the steel truss with more types of loads with different distribution types.

In the paper, the authors used reliability analysis, connected with the FORM method, which can be easily done using special software. In this article, it was the SYSREL program. The most difficult part during system analysis is searching cut-sets, but the author-prepared C++ code enables us to find it effectively.

In summary, it should be said that system reliability analysis for truss elements can be successfully used for a structure under fire influences or in persistent design situations, but there are still some problems to solve and issues to develop. The presented paper is the part of wider research which will be continued in the nearest future. Especially, the authors are going to develop and refine the proposed algorithm. Its usefulness for another structures, including spatial trusses, will be tested.

**Author Contributions:** Conceptualization, K.K. and M.S.; methodology, K.K; software, K.K.; validation, K.K. and M.S.; formal analysis, M.S.; writing—original draft preparation, K.K.; writing—review and editing, M.S.; visualization, K.K.; supervision, M.S. All authors have read and agreed to the published version of the manuscript.

**Funding:** The APC was funded by the program of Minister of Science and Higher Education "Regionalna Inicjatywa Doskonałości (RID)" No. 025/RID/2018/19 of the day 28 December 2018.

**Institutional Review Board Statement:** Not applicable.

**Informed Consent Statement:** Not applicable.

**Data Availability Statement:** Not applicable.

**Acknowledgments:** The authors gratefully acknowledge the contribution of the Scientific Grant Agency of the Slovak Republic under the grant VEGA no. 1/0230/22.

**Conflicts of Interest:** The authors declare no conflict of interest.

## Appendix A

Table A1 includes some information about the analyzed truss (Figures 3 and 4) in the 30th minute of fire duration, including: effects of action, bearing capacities, and reliabilities for individual members. The reliabilities of elements were computed according to Equations (2)–(6) with the assumption that the coefficient of variation of effect of action and bearing capacity are equal to 6% and 10%, respectively.

**Table A1.** Effect of action, bearing capacity, and reliability of analyzed truss elements in 30th minute of fire duration.

| Element No. | 1 | 2 | 3 | 4 | 5 | 6 | 7 | 8 | 9 | 10 | 11 | 12 |
|---|---|---|---|---|---|---|---|---|---|---|---|---|
| Effect of Action (kN) | 133 | 133 | 224 | 224 | 133 | 133 | −167 | −213 | −229 | −229 | −213 | −167 |
| Bearing Capacity (kN) Ncfi/Nbfi | 392 | 392 | 392 | 392 | 392 | 392 | 284 | 1264.30 392 | 392 | 392 | 392 | 284 |
| Reliability (-) | 1 | 1 | 1 | 1 | 1 | 1 | 0.99995 | 0.999993 | 0.99996 | 0.99996 | 0.999993 | 0.99995 |

| Element No. | 13 | 14 | 15 | 16 | 17 | 18 | 19 | 20 | 21 | 22 | 23 |
|---|---|---|---|---|---|---|---|---|---|---|---|
| Effect of Action (kN) | 40 | −11 | 16 | −12 | 40 | −13 | 20 | 100 | 100 | 20 | −13 |
| Bearing Capacity (kN) Ncfi/Nbfi | 66 | 66 | 66 | 66 | 66 | 29 | 251.45 29 | 29 | 29 | 29 | 29 |
| Reliability (-) | 1 | 1 | 1 | 1 | 1 | 0.99999995 | 1 | 0.999999998 | 0.999999998 | 1 | 0.99999995 |

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
