# Peer review of "Fire Safety of Plane Steel Truss according to System Reliability Analysis Combined with FORM Method: The Probabilistic Model and SYSREL Computation"

_applsci, doi:10.3390/app13042647_

Round 1

Reviewer 1 Report

According to the reviewer, the work should be published.
The work deals with the important topic of estimating the reliability of engineering structures. EN standards require such calculations (class CC3), but they do not specify how to perform them.
The paper shows that the use of different probabilistic models leads to different results. The work does not provide ready-made computational algorithms, but it can inspire a discussion on how to include probabilistic methods in engineering calculations.
The calculations were carried out for the structure in fire conditions. Therefore, the analysis performed is of a particularly important nature. However, it should be emphasized that the conclusions have a much broader meaning and do not refer only to the presented case.

The work requires editorial corrections, e.g. numbering of formulas 7.1 and 7.2 (line 168) and in many other places. References to formulas in the text are also incorrect.

Author Response

Dear Sir or Madam,

Thank you very much for the positive review and indicating some editorial mistakes that we overlooked. Of course, they were corrected. Due to your comment, the ‘Conclusion’ part was developed, where authors emphasized that the results are not only limited to the presented case.

Furthermore, we are aware that providing the ready-made algorithm could be beneficial for the scientific community, but it is hardly possible at this moment. First, it is too long to be presented in the manuscript. Second, it is still developing phase. In our opinion, the more beneficial would be discussion during the conference or internship with scientists who would be interested in the subject.

Yours faithfully,

Katarzyna Kubicka,

Milan Sokol

Reviewer 2 Report

The computational algorithm proposed by the authors seems interesting and factually correct, however, according to the reviewer, it should be described in a much more precise manner, with attention to formal details. Firstly, the English language used in the text requires professional proofreading. Secondly, the substantive reasoning should be shortened and thus condensed, so as not to provide generally known information in the text. It should be clearly shown which quantities are treated as deterministic in the applied model and which are random. The buckling coefficient specified for an element is a random value, therefore treating it by the authors in a sample computational model as a deterministic value seems unjustified. The authors do not introduce the fire duration treated as a formal parameter into their model. For this reason, in the proposed approach, they simply determine the reliability of the analysed truss only for a fixed moment of a fire. Nevertheless, this kind of information seems to be cognitively valuable because the performed analysis can be repeated many times by assigning succesive moments of a fire to the calculations. In the presented article, a specific calculation example is analysed. Unfortunately, the results obtained on its basis are neither presented nor discussed in a satisfactory manner. This lack needs to be supplemented with a substantive commentary. External loads applied to the truss nodes should be treated in calculations as random values, with a given mean value and a given coefficient of variation. On this basis, the parameters of the random load effect should be determined. There is no information on analogous parameters describing random coefficients of reduction under fire conditions of the yield strength and also of the modulus of elasticity specified for structural steel. How were they modelled by the authors? In many places in the text, the authors provide imprecise information. For example, the Laplace function used in relation (1) is not a simple cumulative distribution function. There is also no EN 1990-1-2 standard document. In the description of the quantities used in formula (4) it is not specified whether we are talking about the mean values or only about the random values. In all places in the text where the word "whose" is used by the authors, it should be replaced with the word "that". The general conclusion of the reviewer is that the proposed approach is substantively interesting, although its description needs to be formally reworded to be sufficiently precise.

Author Response

Dear Sir or Madam,

Thank you very much for the very detailed and deep review that allowed us to improve the manuscript. Of course, we introduced adjustments to the manuscript based on your comments. But we would also like to answer about some doubts you indicated.

  1. ‘The buckling coefficient specified for an element is a random value due to its dependence on many random values, so treating it by the authors in a sample computation model as a deterministic value seems to be unjustified.’ The buckling coefficient is the function of the random variables, which were defined in the algorithm , so as the results of the computation it is also the random variable. It was mentioned in the article, but maybe not clearly enough, so the comment was added. In addition, the description of the variables was extended to unambiguously indicate which of them are random.
  2. In fact, in the article the results are presented for the fixed moment of a fire, but it is possible to get the results for any change of the moment during the fire. Because the temperature of element is computed in the incremental way, during the computation, for example, in the 30th minute of fire duration, it is easy to get results in any moment between 0-30 minutes.
  3. The aim of the authors was to check the influence of the probabilistic model on the value of the reliability index and to demonstrate that there is a strong need to prepare the appropriate models if the scientific community is interested in application of reliability methods as an alternative for the traditional design based on partial safety factors. The presented paper is the part of scientific work which has been realised since few years and there are still some issue to be solved. In particular, the probabilistic character of the reduction factors for the yield strength and bearing capacity should be tested experimentally. This task is planned to be realized at the autumn of 2023 in Kielce University of Technology. The lack of appropriate knowledge at this moment was the cause why the parameters ky and kE were assumed to be deterministic.
  4. Additional information about the probabilistic character of applied loads has benn introduced in the manuscript.
  5. We have some doubt about the shortening of the substantive reasoning. We intentionally introduce it to make the manuscript available to the wider group of recipients, including those scientists who are not very familiar with reliability theory.

Finally, we would like to inform you that the paper is part of the biggest research we are going to continue in the near future. We are aware there are still some “lacks” of the manuscript, which you fairly notice, like for example the probabilistic character of the reduction factors ky and kE.

We hope that the answer, correction, and improvement of the manuscript have been done carefully and sufficient enough.

Yours faithfully,

Katarzyna Kubicka,

Milan Sokol

Reviewer 3 Report

Authors analysed the reliability of the plane steel truss under fire conditions by system reliability analysis combined with First Order Reliability Method (FORM). The C++ code created enables to prepare the advanced probabilistic model for bearing capacity in selected time of fire duration. The probabilistic model’s significant influence on the reliability indices was well evaluated. The paper can be published after Journal’s style adjustment.

Author Response

Dear Sir or Madam,

Thank you very much for the positive review. We would like to inform you that the manuscript has been improved according to all reviews. There is new version of it available.

Yours faithfully,

Katarzyna Kubicka,

Milan Sokol

Round 2

Reviewer 2 Report

The authors put a lot of work into correcting the existing text. They also greatly improved the English language. Nevertheless, in the opinion of the reviewer, the substantive argument is still written in a rather chaotic manner. The method of assessing the fire safety of a steel truss, proposed by the authors, is undoubtedly original, very interesting and worth publishing. It must, however, be explained to the reader in a more orderly and precise manner. In its current version, a lot of information is repeated many times. Many of the assumptions adopted for the analysis seem to be very problematic and raise objections of the reviewer. Much content can be removed without compromising the quality of formal analysis. The numerical example proposed by the authors may itself be a topic for a separate article, provided that it is presented much more precisely. In conclusion, the reviewed work seems worth publishing, but it is necessary to thoroughly re-edit it. By the way, "safety margin" is not the same as "margin safety" (line 186). Besides, in formula (6) we use the mean value of this margin, while in formula (4) only a random value of it is used. These are generally not the same values.

Author Response

Dear Sir or Madam,

Thank you very much for the review with accurate comments. We did our the best to improve the manuscript according to your suggestions. The section “Materials and methods” was supplied with the figure that presents the algorithm used during the research. We hope this makes the article easier to understand by the reader.

We agree that some of the assumptions are very ‘strong’ and actually they can raise some objections. But we would like to underline that the presented results are part of wider research which combined two topics: reliability analysis and design of the structure under fire conditions. In the following part of the research, the authors are going to resign from the consecutive assumption, but in our opinion it is reasonable to do this gradually.

The numerical example in the manuscript was introduced to present the influence of a probabilistic model on the results obtained from the reliability analysis. The results are obtained from the SYSREL program which computes the reliability according to the methods described in 2.2-2.4 sections.

Detailed information on searching cut-sets was previously described in the cited papers, so the authors decided not to copy the information. As the research is going to be continued it is not impossible to completely re-edit the manuscript, because it will destroy the general idea of research. The aim of the article was to present some possibilities of the proposed algorithm, which will be refined and developed.

Yours faithfully,

Katarzyna Kubicka,

Milan Sokol

Round 3

Reviewer 2 Report

In the reviewer's opinion the current version of the article still contains many problematic theses and assumptions. Nevertheless I propose to allow it for publication, hoping that in the future the algorithm recommended by the authors will be developed in a much more precise way. The mandatory condition for this publication is, however, the formal correction of dependence (6), to which an appropriate mean value (written for example as mean M) is introduced in place of the currently existed random M value.

Author Response

Dear Sir or Madam,

Thank you very much for understanding. The mistake in equation (6)  was corrected. We are sorry, we missed it after your previous review.

Yours sincerely,

Katarzyna Kubicka,

Milan Sokol